# Protein Phosphatase OsPP2C55 Negatively Regulates Abscisic Acid Biosynthesis and Saline–Alkaline Tolerance in Rice

**DOI:** 10.3390/plants14213362

**Published:** 2025-11-03

**Authors:** Gang Zhang, Yi Yang, Yuhan Jing, Mengjiao Xin, Shuxian Shi, Qingshuai Chen, Ke Yao, Mengyu Su, Lijing Wang, Mingyi Jiang

**Affiliations:** 1Institute of Biophysics, Dezhou University, Dezhou 253023, China; yangyi@dzu.edu.cn (Y.Y.); jyh05730@126.com (Y.J.); mjxin0917@126.com (M.X.); ssx202507@126.com (S.S.); qingschen_dzu@dzu.edu.cn (Q.C.); 17794539779@163.com (K.Y.); smy@dzu.edu.cn (M.S.); qwjingjing@163.com (L.W.); 2National Key Laboratory of Crop Genetics and Germplasm Enhancement, College of Life Sciences, Nanjing Agricultural University, Nanjing 210095, China; m_y_jiang@126.com

**Keywords:** ABA biosynthesis, OsPP2C55, OsABA2, rice, saline–alkaline stress

## Abstract

In rice (*Oryza sativa* L.), the short-chain dehydrogenase protein OsABA2 plays a crucial role in regulating abscisic acid (ABA) biosynthesis. However, little is known about the other proteins that interact with OsABA2 to regulate ABA biosynthesis. Using yeast two-hybrid screening, we identified a novel OsABA2 interacting protein OsPP2C55, which contains a serine/threonine phosphatase (family 2C) catalytic domain. The yeast two-hybrid (Y2H) assay and firefly luciferase complementary imaging (LCI) assay confirmed these interactions. Subsequent studies revealed that saline–alkaline stress significantly downregulated *OsPP2C55* gene expression. Meanwhile, we constructed *ospp2c55* CRISPR gene knockout (*ospp2c55*-KO) plants using Agrobacterium genetic transformation. Compared with wild-type plants, *ospp2c55*-KO plants under saline–alkaline stress exhibited significantly elevated OsABA2 protein levels, leading to substantial increases in ABA content. In addition, *ospp2c55*-KO plants demonstrated heightened sensitivity to ABA during seed germination. Moreover, *ospp2c55*-KO plants improved the survival rate and stress-related indices of rice seedlings under saline–alkaline stress, and upregulated the expression of genes related to adversity stress (*OsNCED1*, *OsNCED3*, *OsABA2*, *OsSODCc2*, and *OsCatB*). We found that *OsPP2C55* plays a negative regulatory role in ABA biosynthesis and saline–alkaline stress tolerance in rice.

## 1. Introduction

Soil salinization is a major abiotic stressor affecting the sustainable development of global agriculture [1]. Rice, one of the world’s most important cereal crops, serves as a staple food for over half of humanity. Among cereal crops, rice is the most sensitive to salt stress. Therefore, salt tolerance in rice remains the focal point of scientific research [2]. Saline–alkaline soils are categorized into neutral pH saline soils (rich in NaCl and Na_2_SO_4_) and high pH saline–alkaline soils (containing Na_2_CO_3_ and NaHCO_3_, accounting for approximately 60%). While current research has provided substantial insights into plant salt stress tolerance, there remains a significant lack of understanding regarding plant alkaline stress tolerance (also referred to as saline–alkaline stress) [3]. Saline–alkaline stress has many adverse effects on crops [4,5,6]. On the one hand, osmotic stress, ionic toxicity, and oxidative stress caused by high salt stress not only make it difficult for crop roots to absorb water but also affect the physiological and biochemical functions of crops; excessive accumulation of Na^+^ also damages protein synthesis and affects photosynthesis. On the other hand, high pH causes more serious damage to crops, severely restricting agricultural production [5,7]. Therefore, a systematic study of the molecular mechanisms of plant saline–alkaline tolerance and the exploration of key genes and regulatory networks of saline–alkaline tolerance are of great significance for solving the increasing demand for food and ensuring food security.

Abscisic acid (ABA) plays an indispensable role in the plant stress response. First, it detects environmental stress signals and initiates the downstream mechanisms. When soil salinity increases or osmotic pressure becomes unbalanced, ABA biosynthesis intensifies in the roots, which is then transported through the xylem to the leaves [8,9,10]. This triggers rapid stomatal closure to minimize water loss [11]. Additionally, ABA regulates salt stress tolerance-related gene expression by activating genes encoding vacuole membrane Na^+^/H^+^ antiporters (NHX) and the antioxidant enzymes superoxide dismutase (SOD) and catalase (CAT), thereby enhancing ionic compartmentalization and reactive oxygen species (ROS)-scavenging efficiency to mitigate ion toxicity and oxidative damage [12,13]. Third, ABA induces plants to accumulate osmotic stress substances such as proline and soluble sugars, which elevate cellular osmotic pressure while stabilizing cell membranes [14]. This dual action reduces the enzyme activity disruption caused by salt ions, enabling plants to maintain normal metabolism and growth in saline–alkaline environments. Ultimately, plants develop adaptive mechanisms through ion balance regulation, osmotic homeostasis, antioxidant systems, and molecular signaling pathways to thrive under saline–alkaline conditions [15,16,17]. Therefore, it is important to study the molecular mechanisms underlying the regulation of ABA biosynthesis.

Protein phosphatases are the core enzymes that regulate protein phosphorylation in plants. By removing phosphate groups from target proteins, they work with protein kinases to form a “phosphorylation-dephosphorylation” switch, precisely controlling cellular processes such as signal transduction, metabolic balance, and growth development. This mechanism is crucial in stress adaptation [18]. The PP2C family serves as a key negative regulator of ABA signaling. Under normal conditions, PP2C binds to ABA receptors (PYR/PYL/RCAR) and dephosphorylating downstream SnRK2 kinase to inhibit ABA signaling. Under stressful conditions, ABA induces conformational changes in the PP2C after receptor binding, thereby releasing its inhibitors. This activates SnRK2 to phosphorylate downstream transcription factors, thereby initiating the expression of stress resistance genes [19,20,21]. In wheat, TaPP2C158 can depyrophosphorylate TaSnRK1.1, negatively regulating the TaSnRK1.1-TaAREB3 signaling pathway, thus playing a negative regulatory role in wheat drought resistance [22]. StHAB1, a PP2C-type protein in potatoes, interacts with almost all ABA receptors and plays a negative regulatory role in potato drought resistance [23]. In tomatoes, SlPLL2, a protein phosphatase of the PP2C family, plays a key role in systemin-mediated defense gene expression and enhances plant resistance to insects [24]. The SOS signaling pathway plays an important role in maintaining sodium ion homeostasis. In *Arabidopsis thaliana*, PP2C.D6 and PP2C.D7 have been shown to negatively regulate the SOS signaling pathway through dephosphorylation [25]. In conclusion, the PP2C family plays a crucial role in ABA signal transduction and stress resistance in plants. However, it is not clear whether OsPP2C55 participates in saline–alkaline stress resistance and its molecular mechanism in rice.

Xanthoxin dehydrogenase (Rice OsABA2) is a key gene in the ABA biosynthesis pathway [26]. Previous studies have demonstrated that OsMPK1 and OsbHLH110 interact with OsABA2, and that their phosphorylation modifications and transcriptional regulation play crucial roles in ABA biosynthesis and rice salt stress tolerance [27,28]. However, many unknown interacting proteins have yet to be identified. In this study, we identified a novel OsABA2 interacting protein, OsPP2C55, which participates in rice saline–alkaline stress tolerance by negatively regulating ABA biosynthesis.

## 2. Results

### 2.1. OsPP2C55 Is a Novel Interacting Protein of OsABA2, Which Is Inhibited by Saline–Alkaline Stress

As a key enzyme in the rice abscisic acid (ABA) biosynthesis pathway, the OsABA2 protein interaction network is crucial for deciphering ABA regulatory mechanisms. In this study, we constructed a recombinant pGBKT7-OsABA2 vector using OsABA2 as a bait protein. After transforming the Y2HGold yeast strains, yeast two-hybrid screening was performed using rice cDNA libraries (Y187 strain). Through SD/-TLHA nutritional deficiency medium screening with AbA and X-α-Gal colorimetric verification, the candidate interactor OsPP2C55 was identified, which contains serine/threonine phosphatases, family 2C, catalytic domain (Appendix A). Using WoLF PSORT (http://www.genscript.com/wolf-psort.html, accessed on 30 October 2025), we analyzed the subcellular localization of OSPP2C55, revealing that 60.9% of it is likely to be in the cytoplasm and 30.4% in the nucleus. This is consistent with OsABA2’s subcellular localization [27], suggesting potential spatial interaction between OsPP2C55 and OsABA2. The interaction between OsPP2C55 and OsABA2 in the Y2H assay is shown in Figure 1A. To further validate this interaction, OsPP2C55 and OsABA2 were cloned into pCAMBIA1300-nLUC and pCAMBIA1300-cLUC vectors for tobacco leaf transformation. Using the firefly luciferase complementary imaging system (LCI) assay, fluorescence signals were detected by live imaging after luciferin substrate injection using the firefly luciferase complementary imaging system. The observed fluorescence confirmed the interaction between OsPP2C55 and OsABA2 in plants (Figure 1B), providing a key target for the subsequent analysis of OsABA2-mediated ABA signaling pathways. To investigate the expression patterns of *OsPP2C55* in rice, we first examined tissue-specific expression of *OsPP2C55* using qRT-PCR. The results showed that *OsPP2C55* had the highest expression in the leaves, followed by its expression in the roots, and the lowest expression in the seeds (Figure 1C). We then analyzed *OsPP2C55* expression under saline–alkaline (NaHCO_3_) treatment and found that the expression of *OsPP2C55* was rapidly inhibited by NaHCO_3_ (Figure 1D). Maximum decrease in *OsPP2C55* expression was observed 240 min after treatment with NaHCO_3_. These results indicate that *OsPP2C55* may play a negative role in the response to saline–alkaline stress.

### 2.2. OsPP2C55 Is Involved in the Regulation of Abscission Acid Biosynthesis

To investigate the role of OsPP2C55 in ABA biosynthesis, we constructed a gene-knockout mutant strain of *ospp2c55* and obtained two homozygous lines (Figure 2A). Western blot analysis revealed that the *ospp2c55* knockout mutant showed significantly increased OsABA2 protein levels under saline–alkaline stress (Figure 2B,C). Finally, ABA content measurements demonstrated that the *ospp2c55* knockout mutant showed substantially elevated ABA concentrations under saline–alkaline stress conditions (Figure 2D). Collectively, these findings indicated that the *ospp2c55* knockout mutant upregulated OsABA2 protein expression, leading to a marked increase in ABA levels. This confirms that *OsPP2C55* is a negative regulator of ABA biosynthesis.

### 2.3. OsPP2C55 Reduces ABA Sensitivity in Rice Seed Germination

ABA plays a crucial role in inhibiting seed germination. OsABA2, a key gene in ABA biosynthesis, is a critical component of the ABA signaling pathway. However, the role of OsPP2C55, the interacting protein of OsABA2 in seed germination, remains unclear. To investigate this mechanism, we treated rice seeds with different ABA concentrations. The results showed that in the absence of ABA, both wild-type and *ospp2c55* mutant rice seeds germinated rapidly, with no significant difference in germination rate. When exposed to 1 and 5 μM ABA, *ospp2c55* mutants exhibited enhanced sensitivity to ABA compared to wild-type counterparts (Figure 3). Collectively, these findings demonstrate that *OsPP2C55* exerts a negative effect on ABA-mediated seed germination responses.

### 2.4. Phenotype and Physiological Indexes of Rice ospp2c55-KO Plants Under Saline–Alkaline Stress Tolerance

To investigate the role of *OsPP2C55* in rice under saline–alkaline stress, *ospp2c55*-KO plants were treated with NaHCO_3_ and phenotypic changes were observed. As shown in Figure 4A, after 10 d of NaHCO_3_ treatment followed by 7 d of rehydration, *ospp2c55*-KO plants exhibited a significantly better growth status than wild-type plants. After 7 d of rehydration, approximately 80% of *ospp2c55*-KO plants recovered to normal growth, while only approximately 50% of the wild-type plants showed similar recovery. The survival rate of *ospp2c55*-KO plants was markedly higher than that of wild-type plants (Figure 4B). These results indicate that *OsPP2C55* reduces rice tolerance to saline–alkaline stress.

In plants’ adaptation to environmental stresses, the Na^+^/K^+^ ratio serves as a key regulator governing physiological metabolism, ionic balance, and stress tolerance. By precisely maintaining this balance between intracellular and extracellular Na^+^ and K^+^ across different organelles, plants mitigate adverse effects such as ionic toxicity, osmotic stress, and metabolic disorders, thereby ensuring healthy growth and survival [29]. For plants, maintaining a low Na^+^/K^+^ ratio is a common strategy to resist most adverse conditions; in order to detect whether *OsPP2C55* is involved in maintaining Na^+^/k^+^ homeostasis, we measured the content of Na^+^ and k^+^ in the leaves of rice seedlings under saline–alkaline stress. As shown in Figure 4C, under saline–alkaline stress, compared with wild-type rice, *ospp2c55*-KO plants exhibited relatively lower Na^+^ content and higher K^+^ content (Appendix A), resulting in a relatively lower Na^+^/K^+^ ratio under saline–alkaline stress. These results indicate that *OsPP2C55* negatively regulates Na^+^/K^+^ homeostasis in rice under saline–alkaline stress.

Previous studies have demonstrated that ABA enhances the activity of antioxidant defense enzymes under stress [30]. Under saline–alkaline stress conditions, *ospp2c55*-KO rice seedlings showed increased ABA accumulation. This raises the question of whether OsPP2C55 participates in the regulation of antioxidant defense enzyme activity under saline–alkaline stress. To investigate this, we measured the activities of the antioxidant defense enzymes SOD and CAT in rice seedling leaves under saline–alkaline stress. As shown in Figure 4D,E, under normal conditions, there were no significant differences in the activities of the antioxidant defense enzymes SOD and CAT between *ospp2c55*-KO and wild-type rice seedlings. However, after 24 h of saline–alkaline stress, SOD and CAT enzyme activities in the *ospp2c55*-KO lines were significantly elevated compared to those in the wild-type plants. This indicates that *OsPP2C55* negatively regulates the activity of SOD and CAT under saline–alkaline stress in rice.

Under adverse stress conditions, cell membranes are damaged, leading to increased permeability and conductivity, thereby reducing plant stress resistance. Additionally, plants accumulate reactive oxygen species (ROS) under stressful conditions, which trigger lipid peroxidation. Methylmalonate disulfide (MDA), a key product of this process, indirectly reflects plant stress tolerance. As shown in Figure 4F,G, under normal conditions, *ospp2c55*-KO rice plants showed no difference in electrolyte leakage or MDA content compared with wild-type plants. However, under saline–alkaline stress, *ospp2c55*-KO plants exhibited significantly lower electrolyte leakage and MDA content than their wild-type counterparts. These results demonstrate that *OsPP2C55* negatively regulates saline–alkaline stress tolerance in rice. Based on the above results, we found that *OsPP2C55* negatively regulates the tolerance of rice seedlings to saline–alkaline stress.

### 2.5. OsPP2C55 Participates in the Regulation of Stress-Related Genes

*OsPP2C55* is involved in the regulation of ABA biosynthesis, seed germination, and saline–alkaline stress. To further validate the role of *OsPP2C55* under saline–alkaline stress, we used qRT-PCR to measure the expression levels of stress-related genes, including those involved in ABA biosynthesis, such as *OsNCED1*, *OsNCED3*, *OsABA2*, and *OsABA8ox2*, as well as the antioxidant defense-related genes *OsSODCc2* and *OsCatB*. As an important stress hormone that plays a crucial role in regulating plant stress responses, as shown in Figure 5A–D, compared to WT plants, *ospp2c55*-KO under saline–alkaline stress exhibited significantly upregulated ABA synthesis-related genes, *OsNCED1*, *OsNCED3*, and *OsABA2*, whereas genes involved in ABA degradation pathways, such as *OsABA8ox2*, were significantly downregulated. This further confirms that *OsPP2C55* negatively regulates ABA synthesis and saline–alkaline stress. Additionally, genes encoding antioxidant defense enzymes are important in stress responses. By regulating the synthesis of enzymes such as SOD and CAT, a multi-level, efficient antioxidant defense network is established to maintain the cellular redox balance. As shown in Figure 5E,F, the antioxidant defense enzyme genes *OsSODCc2* and *OsCatB* were significantly upregulated in *ospp2c55*-KO plants under saline–alkaline stress. Collectively, these results further confirmed that *OsPP2C55* negatively regulates saline–alkaline stress tolerance.

## 3. Discussion and Conclusions

Saline–alkaline stress is an important environmental factor that affects crop growth, development, and yield, and its harmful effects include osmotic stress, oxidative damage, ion toxicity, and morphological inhibition [29,31]. The adaptation of plants to saline–alkaline stress conditions is crucial for agricultural production. The plant hormone ABA plays a key role in this process [32]. Known as the “stress hormone,” ABA has attracted significant research attention. Genes involved in ABA biosynthesis and degradation have also been identified in plants. The key ABA biosynthesis genes include *ABA1*, *ABA2*, *NCED*, *AAO*, and *AtBG1/AtBG2*, whereas those responsible for ABA degradation include *CYP707A1*/*CYP707A3* and *AtUGT71B6* [33,34]. Identification of these genes provides a crucial research foundation for further exploration of the molecular mechanisms underlying ABA biosynthesis.

Previous studies have shown that the OsMPK1-OsABA2 and ZmMPK5-ZmABA2 signaling pathways play important roles in the regulation of ABA biosynthesis and stress responses in rice [27,35]. In addition, OsbHLH110 can directly bind to the *OsABA2* promoter to regulate the expression of the *OsABA2* gene, increase endogenous ABA content, and enhance salt stress tolerance in rice [28]. These results suggest that transcriptional regulation and phosphorylation play important roles in ABA biosynthesis. In this study, we identified a new OsABA2-interacting protein OsPP2C55, which contains a PP2C family domain, using yeast two-hybrid screening. Further research revealed that saline–alkaline stress significantly suppressed the expression of *OsPP2C55*, suggesting that *OsPP2C55* may act as a negative regulator under saline–alkaline stress. Additionally, *ospp2c55*-KO rice plants showed enhanced OsABA2 protein levels under saline–alkaline conditions, which helped maintain higher endogenous ABA concentrations. These results suggest that *OsPP2C55* may act as a negative regulator of ABA biosynthesis under saline–alkaline stress.

PP2C belongs to a class of serine/threonine protein phosphatases that play important roles in plant growth, development, and stress resistance [18]. In maize, ZmPP84 can interact with the ZmMEK1 protein, inhibit the kinase activity of ZmMEK1 via dephosphorylation, and inactivate the downstream kinase ZmSIMK1, thus negatively regulating drought stress tolerance in maize [36]. In tomatoes, SlPP2C2 inhibits the transcriptional activity of SlZAT5 at Ser-65 via dephosphorylation and regulates ethylene synthesis, thus affecting the fruit ripening process [37]. In addition, the NB domain of the immune receptor (NLR) Sw-5b can directly interact with PP2C4, thereby relieving the inhibition of SnRK2.3/2.4 by PP2C4 and activating plant antiviral immunity [38]. In cotton, the GhTOPP4aD-GhRAF36 module regulates the Thr124/Ser357 site of the ABA signal component GhABI1 through phosphorylation/dephosphorylation, thus improving salt stress tolerance of cotton [39]. Seed dormancy and germination play important roles in plant growth and development, and the TaPP2C-a5.1-TaPYLs-TaSnRK2.8 module plays an important role in regulating wheat seed dormancy and germination [40]. In this study, *ospp2c55*-KO rice lines showed enhanced sensitivity to ABA. Under saline–alkaline stress, *ospp2c55*-KO plants exhibited relatively lower Na^+^ and higher K^+^ contents, resulting in a relatively lower Na^+^/K^+^ ratio. These results suggest that *OsPP2C55* negatively regulates rice sensitivity to ABA and tolerance to saline–alkaline stress.

Under adverse stress conditions, the cellular structure, physiological metabolism, and molecular homeostasis of rice are severely disrupted. The upregulation of stress-resistance genes plays a crucial role in protecting rice against environmental stress. To further investigate the function of *OsPP2C55* under saline–alkaline stress, we used qRT-PCR to analyze the expression of stress-related genes, including those involved in ABA biosynthesis and antioxidant defense enzymes [41]. Our results demonstrated that *ospp2c55*-KO plants upregulated the ABA biosynthesis-related genes *OsNCED1*, *OsNCED3*, and *OsABA2* under saline–alkaline stress, whereas genes involved in ABA degradation pathways, such as *OsABA8ox2*, were significantly downregulated. Additionally, saline–alkaline stress induces the production of reactive oxygen species (ROS) in cells. The antioxidant enzymes SOD and CAT play vital roles in scavenging ROS and reducing lipid peroxidation damage to cell membranes [31]. Our research findings demonstrate that *ospp2c55*-KO plants exhibit significantly enhanced antioxidant defense capabilities under salinity-alkalinity stress. Notably, both *OsSODCc2* and *OsCatB* were substantially upregulated in the *ospp2c55*-KO plants under these stress conditions. These results further confirm that *OsPP2C55* exerts a negative regulatory activity to improve saline–alkaline stress tolerance.

In conclusion, our previous studies demonstrated that OsMPK1-mediated protein phosphorylation and OsbHLH110-regulated transcription play crucial roles in regulating ABA biosynthesis [27,28]. However, additional regulatory mechanisms of ABA biosynthesis need to be incorporated into this framework [42]. Our study revealed that OsPP2C55 is a novel protein that interacts with OsABA2. Under saline–alkaline stress, *ospp2c55*-KO plants exhibited significantly elevated OsABA2 protein levels, leading to increased endogenous ABA content in rice. Additionally, *ospp2c55*-KO plants under saline–alkaline stress demonstrated enhanced ABA sensitivity, improved seedling survival rates, activated antioxidant defense enzymes, and stress-related gene expression, while the Na^+^/K^+^ ratio and lipid peroxidation levels were reduced under saline–alkaline stress. These findings indicate that *OsPP2C55* may exert negative regulatory effects on both ABA biosynthesis and saline–alkaline stress tolerance. However, further studies are needed to deepen our understanding of ABA biosynthesis.

## 4. Materials and Methods

### 4.1. Plant Materials and Growth Conditions

In this study, rice (*Oryza sativa* L.) was used as the WT plant. The *ospp2c55*-KO plants were constructed by Towin Biotechnology (Wuhan, China) using the CRISPR-cas9 gene knockout technology. Figure 3A shows the identification results for *ospp2c55*-KO plants. Seeds from the T3 homozygous transgenic plants were used for further analysis. Rice seedlings were grown under previously described conditions [43]. The second leaves were collected and used for subsequent experiments.

### 4.2. Y2H Screening and Assay

For Y2H screening, we first cloned the full-length *OsABA2* gene from the rice cDNA library. Using a double digestion strategy, we constructed the pGBKT7-OsABA2 plasmid as a bait. This was then transformed into Yeast strain Y2H Gold. The prey pGADT7 cDNA library was subsequently transformed into Yeast strain Y187. After fusion of the bait and prey strains, the mixture was plated on SD/-TLHA plates with AbA and X-α-gal. Positive clones were identified through PCR amplification followed by sequencing. For the Y2H assay, *OsPP2C55* was cloned into pGADT7 as prey and then transformed into yeast strain Y187. The prey and bait strains were mated by 2 × YPDA, and the mating cultures were spotted on SD/-TLHA plates with AbA and X-α-gal. The plates were incubated at 30 °C for 5–7 days before photographing.

### 4.3. LCI Assay

For the LCI assay, we first constructed the pCAMBIA1300-nLUC-OsPP2C55 and pCAMBIA1300-cLUC-OsABA2 vectors using a double digestion strategy, which were then transformed into *Agrobacterium* GV3101 competent cells. The fused constructs were inoculated onto leaves of *Nicotiana benthamiana*. Following a three-day infiltration culture period, luciferase signal images were captured using a Tanon 5200 multichannel camera (Tanon, Shanghai, China).

### 4.4. Quantitative Real-Time PCR (qRT-PCR) Analysis

Approximately 0.1 g rice tissue (such as leaves, was ground into powder using liquid nitrogen. RNAiso reagent was used to extract total RNA. First-strand cDNA was synthesized from 200 ng of RNA using Hifair Ⅲ 1st Strand cDNA Synthesis SuperMix for qPCR (Yeasen, Shanghai, China). qRT-PCR was performed using 2 × Hieff UNICON Universal Blue qPCR SYBR Green Master Mix (Yeasen) on a CFX96 Real-Time PCR system (Bio-Rad, Hercules, CA, USA). The gene-specific primers used in this study of *OsPP2C55*, *OsNCED1*, *OsNCED3*, *OsABA2*, *OsABA8ox2*, *OsSODCc2* and *OsCatB* are listed in Appendix A.

### 4.5. SDS-PAGE and Immunoblot Analysis

The protein (100 μg) was separated using SDS-PAGE and transferred to a PVDF membrane via electroporation (110 V, 1 h). The PVDF membrane was blocked overnight in 20 mL of 5% skim milk powder solution (prepared with PBST buffer). The next day, the PVDF membrane was washed three times with PBST buffer for 5 min each. Subsequently, 20 mL of 5% skim milk powder solution was added, followed by the addition of anti-ABA2 or anti-actin antibody at 1:1000 dilution. The membranes were then incubated overnight at 4 °C. The next day, after washing with PBST buffer, 20 mL of 5% skim milk powder solution was added, and the secondary antibody was diluted 1:4000. After washing three times with PBST buffer for 3 h, the membrane was observed using a chemiluminescence imaging system and photographed (Tanon, Shanghai, China).

### 4.6. Determination of ABA Content

Rice leaves with uniform growth were selected and washed, the surface moisture was dried using filter paper, and then rapidly frozen in liquid nitrogen before being ground into a powder. The powder (0.5 g) was accurately weighed (three replicates) and transferred to a 10 mL centrifuge tube. Two milliliters of extraction solution was added, vortex-mixed, and extracted under light-sheltered conditions for 12 h at 4 °C. The powder was centrifuged at 12,000 rpm for 20 min, the supernatant removed, and the residue was added with 2 mL of extraction solution. The extraction was repeated once. Both supernatants were combined. The supernatant was filtered through 0.22 μm organic filter membrane and transferred to a Sep-Pak C18 cartridge. The impurities were eluted with 5 mL 5% methanol aqueous solution (0.1% acetic acid), and then ABA was eluted with 5 mL 80% methanol aqueous solution (0.1% acetic acid) to collect the eluates. The eluates were transferred to a rotary evaporator flask and the concentration was reduced under vacuum until nearly dry. Mobile phase was added for dissolution, passed through 0.22 μm filter membrane, and transferred to injection bottle for analysis. Ultra-high-performance liquid chromatography was performed at a detection wavelength of 254 nm. An ABA standard (Sigma-Aldrich, Shanghai, China) was used for quantitative analysis using a standard curve [44].

### 4.7. Phenotype and Stress Tolerance Analysis

In the seed germination experiment, seeds of wild-type (WT) and *ospp2c55*-KO plants were treated with different concentrations of ABA at 0, 1, and 5 μM. After 5 days of treatment, seed germination phenotypes were photographed and germination rates were recorded. For the saline–alkaline stress treatment, ten-day-old rice seedlings were treated with 100 mM NaHCO_3_ for 10 d and then allowed to recover for 7 d. Survival rates were measured and the phenotype was photographed. Additionally, Na^+^ and K^+^ contents, enzyme activities of SOD and CAT, percentage of electrolyte leakage, and MDA content were measured following the previously described methods [41,45,46].

For the detection of Na^+^ and K^+^ contents, rice seedlings were treated with 100 mM NaHCO_3_ for 24 h. Subsequently, the rice leaves were dried at 80 °C until constant weight and ground into powder. We accurately weighed 0.02 g of the sample, added 800 μL nitric acid to a 2 mL centrifuge tube, acidified at 100 °C for 1 h, and then acidified at 80 °C for 6 h. After cooling, we filtered the sample, took 40 μL, diluted with 5% nitric acid to 4 mL, and analyzed the content of Na^+^ and K^+^ ions using inductively coupled plasma mass spectrometry (ICP-OES, Shelton, MA, USA).

For the analysis of SOD and CAT activities, uniformly grown rice seedlings were pre-treated for 2 h to eliminate mechanical damage, and then incubated with 100 mM NaHCO_3_ (control group treated with water) for 24 h. A 0.1 g sample was added to 1 mL of enzyme-active extract, homogenized in liquid nitrogen, and centrifuged at 12,000 rpm for 30 min at 4 °C. The supernatant was collected as crude enzyme solution for subsequent activity assays. The SOD enzyme activity unit (U) is defined as the amount of enzyme that inhibits 50% NBT photochemical reaction, quickly measuring the OD560 value using an enzyme microplate reader. The enzyme activity of CAT is measured in units (U) based on the amount of enzyme that reduces OD240 by 0.1 within 1 min.

For the detection of MDA content, weigh 0.4 g of rice leaves and grind them on ice with 4 mL of 10% TCA (*w*/*v*). Transfer the mixture to a 10 mL centrifuge tube and centrifuge at 12,000 rpm for 10 min at 4 °C. Collect 2 mL of supernatant, add an equal volume of 10% TCA (containing 0.6% *w/v* TBA), mix thoroughly, and boil in water for 15 min. Rapidly cool on ice, and then centrifuge at 12,000 rpm for 10 min at 4 °C. Measure absorbance at 450, 532, and 600 nm using the supernatant. MDA content is calculated as follows: MDA concentration (μM) = 40 × [6.45 (A532 − A600) − O.56A 450)]. MDA content (μmol/g) = MDA concentration (μM) × extraction volume (mL)/fresh weight of plant tissue (g).

For the analysis of the percentage of electrolyte leakage, accurately weigh 0.1 g of rice leaves, chop them, and place in a 50 mL centrifuge tube. Add 10 mL ultrapure water for soaking. After incubating at 30 °C for 2 h, measure the initial conductivity (EC1) using a conductivity meter. Subsequently, boil the centrifuge tube in a water bath for 15 min, cool to room temperature, and measure the final conductivity (EC2). Calculate the relative electrolyte leakage percentage using the following formula: Electrolyte leakage rate (%) = EC1/EC2 × 100.

### 4.8. Statistical Analysis

The experimental data were analyzed using SPSS 21 software. Each experiment was repeated at least three times, and the same letter indicates that the difference was not significant at *p* < 0.05, according to one-way ANOVA (* *p* < 0.05).

### 4.9. Accession Numbers

The sequence data from this study can be found in the Rice Genome Annotation Project (http://rice.uga.edu/, accessed on 30 October 2025) with the following accession numbers: *OsPP2C55*, LOC_Os03g59470; *OsABA2*, LOC_Os03g59610; *OsACT1*, LOC_Os03g50885.

## Figures and Tables

**Figure 1 plants-14-03362-f001:**
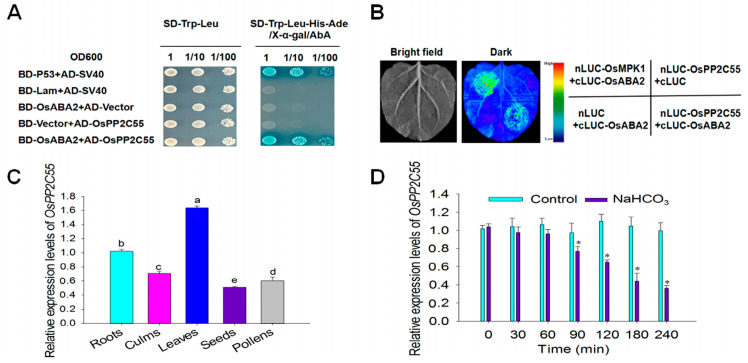
Analysis of OsPP2C55-OsABA2 interaction and *OsPP2C55* expression. (**A**) Interactions between AD-OsPP2C55 and BD-OsABA2 in the Y2H experiment. BD-P53+AD-SV40 cells were used as positive controls. (**B**) The interactions between nLUC-OsPP2C55 and cLUC-OsABA2 were determined using the LCI assay. The nLUC-OsPP2C55 and cLUC-OsABA2 plasmids were transformed into *Agrobacterium* GV3101 competent cells for tobacco infection. After three days of cultivation, tobacco leaves were sprayed with the substrate to capture luciferase (LUC) signal images. Positive control: nLUC-OsMPK1+cLUC-OsABA2. (**C**) The qRT-PCR was used to analyze the expression of *OsPP2C55* in various rice tissues. We collected tissues from different growth stages, including roots, stems, leaves, seeds, and pollen, and extracted total RNA, which was then reverse-transcribed into cDNA. Using actin as a housekeeping gene, specific primers for *OsPP2C55* were designed and used to detect relative expression levels. (**D**) Effect of NaHCO_3_ treatments on the expression of *OsPP2C55* in rice seedling leaves. The 10-day old seedlings were treated with 100 mM NaHCO_3_. Values are means ± SE of three independent experiments; means denoted by the same letter did not differ significantly at *p* < 0.05 according to Duncan’s multiple range test (* *p* < 0.05).

**Figure 2 plants-14-03362-f002:**
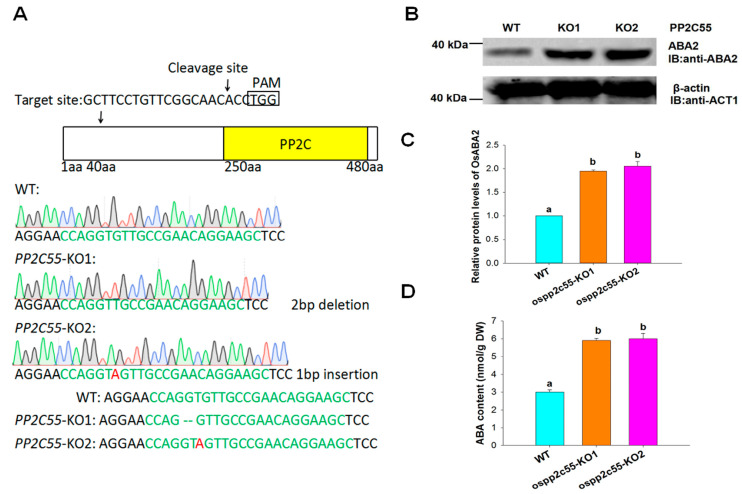
OsPP2C55 negatively regulates abscisic acid biosynthesis by affecting the OsABA2 protein level. (**A**) CRISPR/Cas9-mediated targeted knockout of *ospp2c55* knockout. The arrow indicates the CRISPR/Cas9-mediated targeted knockout sequence of *ospp2c55*, and Sanger sequencing shows the deletion positions of the *ospp2c55* knockout lines. *pp2c55*-KO1: GT deletion. *pp2c55*-KO2: A insertion. PAM: protospacer adjacent motif. (**B**) OsABA2 protein levels were regulated by OsPP2C55. Total protein was extracted from the leaves of *ospp2c55*-KO and WT rice seedlings. (**C**) Quantification of the relative protein levels of OsABA2 from (**B**) by Image J software. The protein level of OsABA2 in wild-type rice was set to 1. (**D**) ABA content in the leaves of WT and *ospp2c55*-KO plants treated with 100 mM NaHCO_3_ for 2 h. In (**C**,**D**), values are means ± SE of three independent experiments; means denoted by the same letter did not differ significantly at *p* < 0.05 according to Duncan’s multiple range test.

**Figure 3 plants-14-03362-f003:**
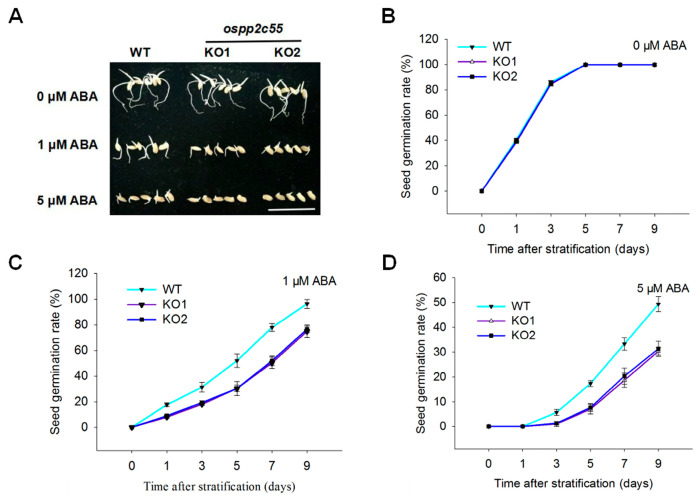
*OsPP2C55* reduces ABA sensitivity in rice seed germination. (**A**) The phenotypes of *ospp2c55*-KO and WT seed germination after 5 days of ABA treatment, scale bar, 5 cm. (**B**–**D**) The germination rate among *ospp2c55*-KO and WT seeds treated under distinct concentrations of ABA. Values are means ± SE of three independent experiments.

**Figure 4 plants-14-03362-f004:**
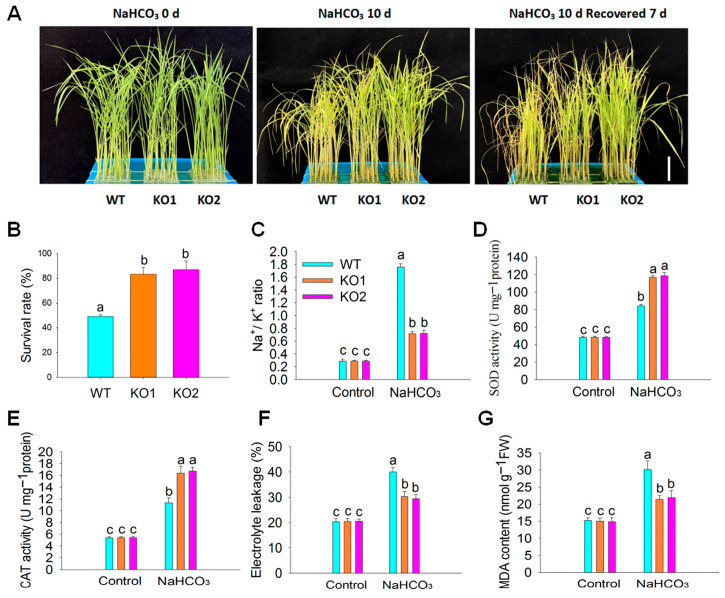
*OsPP2C55* negatively regulates the tolerance of rice to saline–alkaline stress. (**A**) Assessment of phenotypic changes in *ospp2c55*-KO (KO1 and KO2) and WT plants under saline–alkaline stress conditions. The 10-day-old rice seedlings were treated with 100 mM NaHCO_3_ for 10 d, followed by rehydration by re-watering for 7 d, and subsequently photographed. Scale bar, 3.5 cm. (**B**) The survival rate of rice seedlings following exposure to saline–alkaline stress (**A**). (**C**) Na^+^/K^+^ ratios of *ospp2c55*-KO and WT plants under saline–alkaline stress. The Na^+^/K^+^ ratio was calculated using the data shown in Appendix A. (**D**–**G**) The determination of the antioxidant protective enzymes SOD (**D**) and CAT (**E**), percentage of electrolyte leakage (**F**), and MDA (**G**) in rice seedlings under saline–alkaline stress. Ten-day-old seedlings were treated with 100 mM NaHCO_3_ for 24 h, and then leaves were collected for the determination of antioxidant protection enzymes SOD and CAT, MDA, and percentage of electrolyte leakage. Values are means ± SE of three independent experiments; means denoted by the same letter did not differ significantly at *p* < 0.05 according to Duncan’s multiple range test.

**Figure 5 plants-14-03362-f005:**
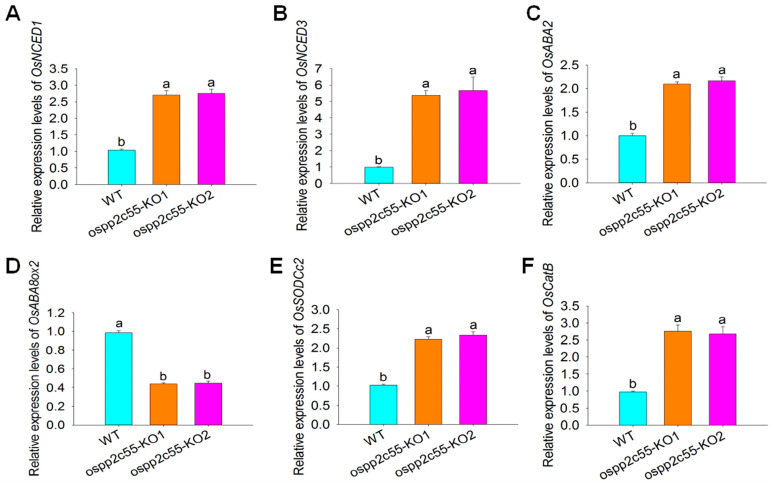
*OsPP2C55* negatively regulates the expression of stress-related genes. Expression of stress-related genes *OsNCED1* (**A**), *OsNCED3* (**B**), *OsABA2* (**C**), *OsABA8ox2* (**D**), *OsSODCc2* (**E**), and *OsCatB* (**F**) in *ospp2c55*-KO and WT rice under saline–alkaline stress conditions. The 10-day-old seedlings were treated with 100 mM NaHCO_3_ for 4 h and leaves were collected for gene expression detection. Values are means ± SE of three independent experiments; different letters indicate that the difference is significant at the *p* < 0.05 level according to one-way ANOVA.

## Data Availability

All data included in this study are available from the corresponding author upon request.

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
