# Peer review of "Protein Phosphatase OsPP2C55 Negatively Regulates Abscisic Acid Biosynthesis and Saline–Alkaline Tolerance in Rice"

_plants, 2025, doi:10.3390/plants14213362_

Round 1

Reviewer 1 Report

Comments and Suggestions for Authors

General Comments

The manuscript of Zhang et al. presents a well-structured and timely study on the role of OsPP2C55 in ABA regulation and saline-alkaline tolerance and germination in rice. The experiments are generally well-designed; however, several minor but important revisions are needed to clarify regarding methodological details, reproducibility and presentation.

Detailed Comments

  • Lines 27-28: The phrase ‘It has been suggested that ...’ is ambiguous. As it is authors conclusion, it should be emphasized as their own suggestion.
  • Paragraph 2.1: authors verified the interaction between OsPP2C55 and OsABA2. Clarify whether the serine/threonine phosphate domain participates in the interaction, or if there is a conserved motif/domain with the ability to interact with OsABA2. If 3D protein structures are available, a docking analysis could strengthen the finding.
  • In Supplementary Figure S1, replace ‘analyze’ with ‘depict’ or ‘present’.
  • Figure 1B: Explain why you used nLUC-OsMPK1 construct and specify the negative controls used. Lines 121-123 should be moved from the figure legend to the respective result text.
  • Figure 2, 3C, D and 6: Provide information about statistical analysis: indicate significant levels (P ≤ 0.05), clarify whether error bars represent SD or SEM, include the number of biological and technical replicates.
  • Paragraph 2.3: Include validation of the knockouts in the Supplementary Materials (qPCR confirmation?) Clarify knockout type (deletion/insertion) and reconcile text with Figure 3, which shows two knockouts.
  • Authors should add a paragraph at Material and Methods explaining how they construct the vectors used at Y2H, CRISP-R, pCambia.
  • Authors should add how they quantify the relative protein levels of OsABA2 at Method & Material section.
  • Figure 3A: Define ‘PAM’ and specify mutation type in line 154
  • Paragraph 2.5: include the concentration of NaHCO3 at text not only in the figure legend. The methods used for Na+/K+ ratio, CAT, SOD, electrolyte leakage and MDA should be mentioned
  • Material and methods: Expand to include (a) vector construction details for Y2H, CRISP, pCampia; (b) Methods more extended for Na+/K+ ratio, CAT and SOD, electrolyte leakage and MDA; (c) Protein purification protocol followed
  • Discussion: Remove figure references

Reviewer 2 Report

Comments and Suggestions for Authors

Dear Authors,

I have carefully reviewed your manuscript entitled “Protein Phosphatase OsPP2C55 Negatively Regulates Abscisic Acid Biosynthesis and Saline-Alkaline Tolerance in Rice.” The study addresses an important topic in plant molecular biology, particularly the regulation of ABA biosynthesis and saline-alkaline stress tolerance in rice, a relevant area for crop improvement. While the research is interesting, several aspects of the manuscript require improvement to enhance scientific clarity, reproducibility, and impact.

Novelty and Rationale:

  1. The manuscript identifies OsPP2C55 as a new OsABA2-interacting protein, which is novel. However, the scientific rationale should be strengthened.
  2. Clearly explain why OsPP2C55 was selected for investigation among the many PP2C family members in rice.
  3. Include a brief phylogenetic or sequence comparison to show how OsPP2C55 relates to other PP2Cs known to regulate ABA or stress signaling.

Experimental Design and Controls:

  1. Yeast two-hybrid and LCI validation: While interaction assays were done, there is no evidence showing subcellular localization or co-immunoprecipitation (Co-IP) in rice. Including one of these would confirm interaction under native conditions.
  2. Expression analysis (Figure 2): The description lacks biological replicates, error bars, or statistical details. Clarify how many replicates were used and include standard deviation or standard error.
  3. Knockout verification (Figure 3): The CRISPR/Cas9 mutagenesis should show mutation sequences, off-target verification, and ideally segregation of the Cas9 transgene.
  4. Complementation test: To confirm that the observed phenotype results specifically from OsPP2C55 loss, a complementation line (OsPP2C55 reintroduction) should be analyzed. This would strengthen causality.
  5. ABA quantification: Include a description of analytical methods (e.g., LC-MS/MS or ELISA) and normalization. Indicate if calibration curves or internal standards were used.

Data Presentation and Statistical Analysis:

  1. Figures should include error bars, statistical tests, and significance indicators (e.g., p < 0.05). Many current figures lack these details, limiting data interpretation.
  2. The units for enzyme activity (SOD, CAT) and ABA concentration should be clearly stated.
  3. Ensure consistent figure labeling, some figures (e.g., Figure 5) lack clear legends or are missing sub-panel details (e.g., D–G).
  4. Include sample size (n) and statistical test used in each figure legend.

Mechanistic Insights:

  1. The discussion should go beyond descriptive results. Consider including a working model figure summarizing how OsPP2C55 modulates ABA biosynthesis and saline-alkaline tolerance.
  2. It remains unclear how OsPP2C55 mechanistically suppresses OsABA2 protein accumulation, does it act through direct dephosphorylation or transcriptional regulation? Adding in vitro phosphatase activity assays or phosphorylation analysis (e.g., Phos-tag SDS-PAGE) would greatly strengthen the study.
  3. The authors repeatedly state that OsPP2C55 “negatively regulates ABA biosynthesis,” but there is no direct biochemical evidence of OsABA2 dephosphorylation. This claim should be toned down or supported with additional data.

Discussion and Literature Integration:

  1. The discussion should be expanded to include comparisons with other PP2Csthat influence ABA pathways (e.g., OsPP2C30, OsPP108). This would place the work in a broader biological context.
  2. Some statements are repetitive (e.g., lines 252–277 restate results without new interpretation). Streamline the discussion to focus on mechanistic insights and implications for breeding.
  3. Clarify whether the observed phenotypes are specific to saline-alkaline stressor extend to neutral salt stress (NaCl). Comparative stress data would add depth.

Writing and Structure:

  1. The manuscript is generally clear, but grammar and sentence structure should be improved throughout (such as: “it was found to OsPP2C55 negatively regulates” → “we found that OsPP2C55 negatively regulates”).
  2. The Abstract should briefly mention the methodological approaches (e.g., Y2H, LCI, CRISPR) and clearly state the novel finding in one sentence.
  3. The Introduction could be shortened to reduce background on general salt stress and focus more on the knowledge gap regarding ABA regulatory proteins.
  4. Avoid redundancy in the Results and Discussion sections, some content overlaps.

Minor Technical Issues:

  1. Correct minor formatting issues in author affiliations (extra commas and inconsistent punctuation).
  2. Ensure all references are properly cited and formatted, and that reference numbers correspond to the correct studies.
  3. Ensure figure numbering matches text citations.
  4. Define all abbreviations (e.g., ROS, MDA, Na⁺/K⁺ ratio) upon first use.

Future Perspectives:

  1. To increase the paper’s impact, the authors could explore whether overexpression of OsPP2C55 causes hypersensitivity to saline-alkaline stress, providing the reverse evidence of its negative regulatory role.
  2. Adding transcriptome or proteome data comparing WT and ospp2c55 mutants under stress could reveal downstream signaling components.

Round 2

Reviewer 2 Report

Comments and Suggestions for Authors

The authors addressed all my concerns in a proper way. 
